# Fraxetin Interacts Additively with Cisplatin and Mitoxantrone, Antagonistically with Docetaxel in Various Human Melanoma Cell Lines—An Isobolographic Analysis

**DOI:** 10.3390/ijms24010212

**Published:** 2022-12-22

**Authors:** Paula Wróblewska-Łuczka, Aneta Grabarska, Agnieszka Góralczyk, Paweł Marzęda, Jarogniew J. Łuszczki

**Affiliations:** 1Department of Occupational Medicine, Medical University of Lublin, ul. Jaczewskiego 8b, 20-090 Lublin, Poland; 2Department of Biochemistry and Molecular Biology, Medical University of Lublin, ul. Chodźki 1, 20-093 Lublin, Poland

**Keywords:** melanoma, coumarins, fraxetin, scoparone, cisplatin, mitoxantrone, docetaxel

## Abstract

Malignant melanoma is a skin cancer characterized by rapid development, poor prognosis and high mortality. Due to the frequent drug resistance and/or early metastases in melanoma, new therapeutic methods are urgently needed. The study aimed at assessing the cytotoxic and antiproliferative effects of scoparone and fraxetin in vitro, when used alone and in combination with three cytostatics: cisplatin, mitoxantrone, and docetaxel in four human melanoma cell lines. Our experiments showed that scoparone in the concentration range tested up to 200 µM had no significant effect on the viability of human malignant melanoma (therefore, it was not possible to evaluate it in combination with other cytostatics), while fraxetin inhibited cell proliferation with IC_50_ doses in the range of 32.42–73.16 µM, depending on the cell line. Isobolographic analysis allowed for the assessment of the interactions between the studied compounds. Importantly, fraxetin was not cytotoxic to normal keratinocytes (HaCaT) and melanocytes (HEMa-LP), although it slightly inhibited their viability at high concentrations. The combination of fraxetin with cisplatin and mitoxantrone showed the additive interaction, which seems to be a promising direction in melanoma therapy. Unfortunately, the combination of fraxetin with docetaxel may not be beneficial due to the antagonistic antiproliferative effect of both drugs used in the mixture.

## 1. Introduction

Malignant melanomas are neoplasms derived from melanocytes and are characterized by early metastasis, rapid development, poor prognosis, and high mortality [1]. The morbidity and mortality of melanoma is increasing worldwide. The standard treatment for melanoma is surgical excision, but sometimes adjuvant (postoperative) treatment is used to prevent recurrence if metastasis is found. The most common adjuvant cancer treatments in melanoma are radiotherapy and chemotherapy [2]. Despite extensive clinical trials, treatment options for this metastatic disease are limited, and melanoma is considered to be one of the most chemotherapy-resistant cancers [3].

Among the numerous chemotherapeutic agents used, we can distinguish mitoxantrone (MTX), docetaxel (DOCX), or cisplatin (CDDP). Mitoxantrone is a synthetic anthraquinone derivative with an anti-cancer effect that inhibits DNA synthesis and transcription. MTX binds to nucleic acids via intercalation and induces apoptosis in both protein-bound DNA and DNA unrelated to the DNA chain breakage site [4]. MTX has two main absorption peaks at wavelengths 610 nm and 660 nm, for which light transmittance through the skin is relatively high; therefore, it is an extremely effective photosensitizer that can mediate the death of human cancer cells at low concentrations with little exposure to light [5]. MTX is only used intravenously; however, it can cause a number of side effects, including bone marrow suppression and cardiotoxicity [6,7].

Docetaxel (DOCX) is one of the taxanes, a diterpene alkaloid, and it belongs to tubule stabilizing drugs [8]. The stabilization process takes place mainly in the mechanism of preventing depolymerization. The drug promotes tubulin polymerization and stabilizes the polymer [9], which is then unable to shorten, and as a result, the cell is unable to divide while maintaining chromosome fidelity. There are three possible consequences—cell death in mitosis, abnormal mitosis, or mitotic slippage [10]. Docetaxel is used in anti-cancer therapy for breast, ovarian, cervical, prostate, bladder, esophagus, stomach, head and neck cancer, non-small cell lung cancer, and Kaposi’s sarcoma [9,10]. In combination with other drugs, it is being tested in the treatment of melanoma [11,12].

Cisplatin (CDDP) is one of the most powerful and widely used anti-cancer drugs that is clinically effective in the treatment of a variety of solid tumors, including ovarian cancer, adrenal cortex cancer, and malignant melanoma [13]. Cisplatin is used in combination with chemotherapy in the first- and second-line treatment of melanoma [14]. The main cytotoxic activity of cisplatin is mediated by its interaction with DNA to form platinum-DNA adducts, leading to DNA damage and induction of apoptosis [15]. Due to the increasing resistance to chemotherapeutic agents (in the case of CDDP, it is based on the reduction of intracellular accumulation of drugs in cancer cells [16,17]), other methods and chemical compounds with anti-cancer potential are sought by researchers worldwide.

Such compounds may be, e.g., coumarins, which are a large class of natural phenolic compounds of plant origin. Coumarins have a variety of pharmacological effects, including antithrombotic, anti-inflammatory, antibacterial, and especially antitumor, antioxidant, and neuroprotective effects [18,19,20]. In recent years, coumarins have become the main topic of anti-cancer drug design and discovery [21,22].

One of the simple coumarins is scoparone (6,7-dimethoxycoumarin), which has been purified from the Chinese herb *Artemisia scoparia* and has been shown to relax smooth muscles, reduce total cholesterol and triglycerides, and delay specific pathomorphological changes in rabbits with hypercholesterolaemia and diabetes. Various properties of scoparone, including the ability to scavenge reactive oxygen species, inhibition of tyrosine kinases, and enhancement of prostaglandin production, have been suggested to account for these findings [23,24,25]. Another interesting property of scoparone is its protective activity against hyperbilirubinemia, which is the reason why it is considered a potential drug for various liver diseases [26,27], and has traditionally been used in Asia to treat neonatal jaundice [28,29]. In addition, scoparone has anti-inflammatory, antioxidant, anti-apoptotic, and anti-fibrotic properties. It seems that its anti-inflammatory effect is due to the inhibition of the transcriptional activity of the nuclear factor kappaB (NF-κB) [27].

Fraxetin (7,8-dihydroxy-6-methoxycoumarin) is a coumarin originally isolated from the plants *Fraxinus bungeana* and *Fraxinus rhynchophylla*. It shows many biological activities, including antibacterial, anti-inflammatory, neuroprotective [30,31,32,33], and antioxidant [34,35]. Fraxetin as a naturally-occurring coumarin is widely available and relatively cheap and is known to have no serious side effects [31].

## 2. Results

### 2.1. Cell Viability Test

FRX, SCP, CDDP, MTX, and DOCX inhibited the viability of human melanoma cell lines, both primary (FM55P, A375) and metastatic (FM55M2, SK-MEL28), in a concentration-dependent manner when applied separately. Incubation of FM55P, FM55M2, A375, and SK-MEL 28 cells with increasing concentrations of FRX resulted in a reduction in melanoma cell viability (Figure 1). Scoparone (SCP) showed a very weak activity limiting the viability of human melanoma cells; therefore, it was omitted in further experiments (Figure 2). The viability of the keratinocyte cell line (HaCaT) and melanocyte cell line (HEMa-LP) was only slightly inhibited at high FRX concentrations. The final concentration of dimethyl sulfoxide (DMSO) in the culture medium used to dissolve the FRX, SCP, MTX, and DOCX did not exceed 0.1% and did not affect cell viability and cell membrane integrity. The significant inhibition of the viability of melanoma lines confirms the specific effect of FRX on melanoma cells. The results are presented as mean ± SEM at each concentration.

### 2.2. Cytotoxicity of Fraxetin—LDH Test Result

Cytotoxicity of fraxetin (FRX) to normal human keratinocytes (HaCaT), normal human melanocytes (HEMa-LP), and malignant melanoma cells (FM55P, FM55M2, A375, and SK-MEL 28) was measured by LDH assay. The LDH test allowed for the assessment of the release of lactate dehydrogenase into the medium, which indicates damage to the cell membrane and cell death [36]. In our experiment, a significant leakage of LDH was observed in the cell lines FM55M2 and SK-MEL28 treated with FRX in the concentration range of 20–200 µM. For the A375 and FM55P cell lines, significant leakage of LDH under the treatment with FRX was observed in the concentration range of above 40 µM and 50 µM, respectively. An important issue is that fraxetin (FRX) did not exert a cytotoxic effect on normal human keratinocyte cells HaCaT and normal human melanocyte cells (HEMa-LP). The amount of LDH released by these cells was significantly lower compared with the melanoma cell lines (Figure 3).

### 2.3. Cell Proliferation Test

Fraxetin (FRX) administered alone dose-dependently reduced the proliferation of FM55P, FM55M2, A375, and SK-MEL 28 melanoma cells. The treatment of all the investigated cancer cells with increasing concentrations of FRX reduced DNA synthesis which was evaluated by measuring BrdU (5-bromo-2′-deoxyuridine) incorporation into cellular DNA in proliferating cells (Figure 4). As presented herein, FRX inhibited the proliferation of all tested melanoma cell lines. FRX in the highest tested concentration of 200 µM inhibited the proliferation of normal human keratinocytes (HaCaT) and melanocytes (HEMa-LP) in approx. 50% (Figure 4).

### 2.4. Isobolographic Analysis of Interactions

Next, we analyzed the concentration-dependent inhibitory effects of CDDP, MTX, and DOCX on melanoma cells to find out if their combinations with FRX could enhance the antitumor effects of these chemotherapeutic agents. The equations of log-probit concentration–response relationship curves (CRRC) for the studied drugs when used alone and for the combinations of CDDP with FRX, MTX with FRX, and DOCX with FRX (Figure 5) allowed for the determination of the median inhibitory concentrations (IC_50_ values ± SEM). The experimentally derived median inhibitory concentration (IC_50_) values for FRX were: 32.42 ± 4.21 µM for FM55P, 46.04 ± 4.17 µM for FM55M2, 44.03 ± 12.02 µM for A375, and 73.16 ± 7.38 µM for SK-MEL 28, respectively. The IC_50_ values for scoparone (SCP) and fraxetin (FRX) are shown in Table 1.

The experimentally derived mean inhibitory concentration (IC_50_) ranging from 1.3 to 3.3 µM for CDDP, from 0.03 to 1.7 µM for MTX, and from 1.27 to 15.87 nM for DOCX were mentioned in previous experiments [37,38].

The test for parallelism confirmed that the experimentally determined concentration–effect lines for FRX and CDDP, FRX and MTX, and FRX and DOCX (administered alone) are mutually non-parallel to each other in FM55P, FM55M2, A375, and SK-MEL28 cell lines. The exception is the combination of FRX with MTX for the A375 line (Figure 5C’), where the parallelism test of the curves showed that the curves are parallel to each other.

Isobolographic analysis of the interaction between FRX and CDDP at the fixed ratio of 1:1 showed additivity in the studied melanoma cell lines (FM55P, FM55M2, and SK-MEL28) (Figure 6A,B,D, Table 2) and additivity with a tendency to synergy for the A375 cell line (Figure 6C, Table 2).

The combination of FRX and MTX at the fixed ratio of 1:1 showed additive interactions for FM55P, A375, and SK-MEL28 (Figure 6A’,C’,D’, Table 2, Table 3) and synergy between FRX and MTX for the FM55M2 cells (Figure 6B’, Table 2).

In contrast, the combination of FRX with DOCX at the fixed ratio of 1:1 produced antagonistic interaction in the A375 cell line and additivity with a tendency to antagonistic interaction in other melanoma cell lines tested (Figure 6A’’,B’’,C’’,D’’, Table 2).

## 3. Discussion

Malignant melanoma is a global problem due to the steady increase in the incidence of the disease worldwide. In 2020, nearly 325,000 cases of melanoma were diagnosed in patients globally and nearly 58,000 people died [1]. It is found mainly among the Caucasian population. People with fair complexion, light colored eyes, and red or blonde hair are particularly vulnerable [39]. The diagnosis of melanoma is important, as well as prompt treatment. Currently, surgery is the treatment of choice. At the moment, there are also no unambiguous indications for adjuvant treatment after surgery, as each case should be considered individually. Classical complementary methods are radiotherapy and chemotherapy, but they do not always bring the desired effect; therefore, new treatment options are sought. There has been radical progress in treatment, especially in patients with an inoperable tumor and existing metastases. Important advanced methods, used in recent years, are immunotherapy and targeted therapy [40]. Despite the progress in the treatment of melanoma, it is necessary to conduct further research and search for new compounds in order to optimize the therapy.

Scoparone, one of the ingredients of herbal Chinese medicine, has a number of pharmacological effects; therefore, it is used in the treatment of many diseases. Its action depends on the absorption and metabolism of this compound. It has been shown that, in human liver microsomes, scoparone is oxidized to isoscopoletin via 6-O-demethylation. The oxidation rate is about 0.2–0.4 µmol/(minx g of protein) and is related to the functioning of CYP2A13, which shows the highest rate of oxidation. Scoparone and its derivatives are completely excreted in the urine within 24 h [41]. Scoparone has been shown to have anti-tumor properties by inhibiting the proliferation of various tumor cell lines. The use of scoparone for 72 h significantly inhibited the proliferation of DU145 prostate cancer cells with IC_50_ = 41.3 µM/L, although in the case of the other PC-3 prostate cancer cell line its inhibitory effect was small. Nevertheless, both cell lines showed reduced proliferation at higher scoparone concentrations (>100 µM/L). In addition, researchers found that high doses of scoparone inhibited the proliferation of breast cancer cell lines (MCF-7 and MDA-MB-231), cervical cancer (HeLa), hepatocellular carcinoma (HepG2 and Hep3B), and colon cancer (HCT-15, HCT-116, and HT-29). In the DU145 prostate cancer line, against which scoparone was the most potent, it was due to cell cycle arrest in the G1 phase without inducing apoptosis. Scoparone inhibited the transcriptional activity of STAT3 and thus suppressed the transcription of oncogenic STAT3 target genes, leading to inhibition of the growth of prostate cancer cells [42].

In the case of studies on scoparone in relation to melanoma, it was shown that this compound at a concentration of 50 µg/mL caused almost an 8.5-fold increase in cell melanin content compared with control cells but had no effect on the proliferation of murine B16 melanoma cells [43]. Our experiments confirm that scoparone in the tested concentration range of up to 200 µM (obtainable taking into account the solubility of this compound in DMSO) did not have a significant effect on the viability of the four lines of human malignant melanoma (FM55P, FM55M2, A375, and SK-MEL28).

Fraxetin seems to be a much more promising natural compound in anti-cancer therapy. It was shown that fraxetin inhibited the proliferation of colon adenocarcinoma cells (cell lines: HCT116 and DLD-1). Its action was related to the arrest of the cell cycle in the S phase and the induction of cell apoptosis. Decreased levels of p-JAK2 after fraxetin application were observed via the JAK2/STAT3 signaling pathway. Additionally, fraxetin was characterized by low toxicity [44]. Numerous studies confirm that fraxetin induces apoptosis of neoplastic cells and may also inhibit the metastasis of some tumors. Fraxetin facilitated apoptosis in MCF-7 breast cancer cells by upregulating the expression of Fas, FasL, and Bax. The inhibitory effect of fraxetin on the proliferation of MCF-7 cells after 48 h of incubation with fraxetin at a dose of 40 µM was approximately 60% [45]. The IC_50_ doses obtained in this study in the range of 32.42–73.16 µM are comparable. Zhand et al. showed that fraxetin also inhibits the growth of non-small cell lung cancer cells with IC_50_ doses of 20.12 µM and 22.45 µM for HCC827 and H1650 lines, respectively. Even at a high concentration of fraxetin (100 µM), it did not show any inhibition of the growth of normal lung cells. Fraxetin inhibited IL-6-induced phosphorylation of tyrosine residue (Tyr705) signal transducer and activator of transcription 3 (STAT3). FRX interacts with STAT3 through hydrogen bounding and hydrophobic interaction [33]. The antiproliferative effects of fraxetin were also observed against endometrial cancer, line RL95-2. In this case, FRX also evoked apoptosis and inhibited mitochondrial oxidation, promoting anaerobic cellular metabolism. Increased AMPK phosphorylation was observed in RL95-2 cells, accompanied by inhibition of mTOR function. This suggests that the mechanism of FRX-induced mitochondrial apoptosis was associated with the specific energy metabolism of tumor cells [46]. In addition, FRX inhibits the proliferation of LM8 osteosarcoma cells in vitro and tumor growth in mice (at a dose of 10 mg/kg) implanted with LM8 cells in vivo. In addition, fraxetin inhibited metastasis to the lungs and liver. Fraxetin (50–100 µM) inhibited the production of IL-10, monocyte chemotactant protein (MCP)-1, and TGF-β during M2 macrophage differentiation by reducing STAT3 phosphorylation without affecting STAT3 expression. These results suggest that the anti-tumor and anti-metastatic effects of FRX may be due to the regulated activation of TAM by differentiating M2 macrophages in the tumor microenvironment [47]. Fraxetin (in doses up to 50 µM) inhibits the growth of hepatocellular carcinoma, which was demonstrated on the Huh7 and Hep3B cell lines. The mechanism of action was associated with cell cycle arrest, induction of apoptosis, depolarization of the mitochondrial membrane, and increased production of reactive oxygen species. Fraxetin did not affect the phosphorylation of kinase 1/2 regulated by extracellular signals, but it decreased JNK and phosphoinositide 3-kinase signaling. Additionally, the inhibitors of mitogen- and fraxetin-activated protein kinases showed a synergistic antiproliferative effect against hepatocellular carcinoma cells [48].

Fraxetin, apart from its anti-cancer properties, has anti-inflammatory, hepatoprotective, and anti-fibrotic properties. An interesting feature of fraxetin is its antioxidant effect [33,48]. Experiments on human SH-SY5Y dopaminergic cells showed that fraxetin (at a dose of 100 µM) protected against reactive oxygen species (ROS), restored the glutathione redox index after rotenone challenge, and reduced lipid peroxidation. These results suggest that natural antioxidants such as fraxetin can prevent rotenone-induced or oxidative-stress-induced apoptotic dopaminergic cell death, which is of great importance in the prevention of Parkinson’s disease [30].

Generally, coumarins are characterized by numerous biological effects, some of which are used in medicine. There are many reports of their anti-cancer activity, but each tumor responds to specific coumarins differently [49]. For example, it has been shown that osthole inhibits the growth of the rhabdomyosarcoma cell line, and in combination with cisplatin (CDDP) it has an additive effect [50]. The coumarins (including xanthotoxin, bergapten, and isopimpinelin) present in plant extracts showed photocytotoxicity to C32 melanoma cells. They induced upregulation of apoptotic signals such as the cleavage of BAX and PARP and increased the upregulation of the p21 protein in the presence of UVA radiation [51]. Another coumarin, umbelliferone, proved to be an effective inhibitor of carbonic anhydrase, modifying the intracellular pH of A375 melanoma cells; it did not induce apoptosis itself, but sensitized cells to dacarbazine [52]. This confirms the need to search for natural compounds, including coumarins and combinations with cytostatics, which would allow for a better treatment of melanoma, especially drug-resistant melanoma. It was shown that other coumarins, such as osthole, xanthotoxin, xanthotoxol, isopimpinellin, and imperatorin, inhibit the growth of melanoma FM55P and FM55M2 cell lines, but the simple coumarin, which is osthole, was characterized by the best action, and the IC_50_ dose was about 67.25–89.58 µM depending on the cell line types. The experiments also confirmed that the combination of the most preferred osthole with cisplatin gave an additive and synergistic interaction depending on the melanoma cell line tested. The combination of CDDP with xanthotoxol and imperatorin caused an additive interaction, while the combination of CDDP with xanthotoxin and isopimpinellin, which was not very effective against melanoma, resulted in an antagonistic interaction [22]. Our experiments confirm that some coumarins, such as in this case, scoparone, show a slight inhibition of melanoma growth, while fraxetin, which like osthole is a simple coumarin, showed greater effectiveness in inhibiting the viability of four human melanoma cell lines. In addition, combining FRX with CDDP or MTX allowed for obtainment of the additive interaction. Additive interaction was observed for the combination of taxanes such as paclitaxel and docetaxel with the naturally derived compound betulinic acid [38]. Our preclinical in vitro experience has shown that fraxetin should not be combined with docetaxel because in combination the drugs mutually suppress their effects, which may weaken the cancer therapy. Different types of interactions may in this case result from the chemical properties of the compounds (betulinic acid is a terpenoid and fraxetin—coumarin), or the mechanisms of action of the combination.

## 4. Materials and Methods

### 4.1. Cell Culture

Two cell lines of primary melanoma, A375 and FM55P, and two lines of metastatic melanoma, SK-MEL28 and FM55M2, were used in the experiments. A375 and SK-MEL28 cell lines were from the American Type Culture Collection (ATCC, Manassas, Virginia, USA), and FM55P and FM55M2 lines were from the European Collection of Cell Cultures (ECACC, Salisbury, UK). The growth conditions of the culture were described earlier [22,38].

### 4.2. Drugs

Cisplatin (CDDP—Sigma-Aldrich, St. Louis, MO, USA) was dissolved in phosphate buffered saline (PBS) with Ca^2+^ and Mg^2+^. Mitoxantrone (MTX—Sigma-Aldrich) and docetaxel (DOCX—Sigma-Aldrich) were dissolved in DMSO as stock solutions. The examined coumarins, fraxetin (FRX) and scoparone (SCP)—(all from Sigma–Aldrich), were dissolved in DMSO as stock solutions. The drugs were dissolved to the respective concentration with culture medium before their use. PBS and DMSO had no effect on cell proliferation.

### 4.3. Cell Viability Assessment

Cell viability was determined using the MTT assay. FM55P, FM55M2, A375, and SK-MEL 28 cells were placed on 96-well plates (Nunc, Roskilde, Denmark) at a density of 2 × 10^4^ cells/mL, 2 × 10^4^ cells/mL, 2 × 10^4^ cells/mL, and 3 × 10^4^ cells/mL, respectively. The next day, the culture medium was removed and cells were exposed to serial dilutions of FRX, SCP, CDDP, MTX, and DOCX, in fresh culture medium. The next steps of the MTT test have been described by us earlier [37,38]. Each treatment was performed in triplicate and each experiment was repeated 3 times.

### 4.4. Cell Proliferation Assay

The cell proliferation BrdU test was performed by the ELISA method using a kit (Roche Diagnostics, Mannheim, Germany). Optimized amounts of cells (as described above) were placed on a 96-well plate (Nunc) (100 µL/well). On the next day, the cancer cells were treated with increased concentrations of FRX for 48 h, followed by 10 µL/well BrdU Labeling Solution (100 µM), and then cells were reincubated for an additional 24 h at 37 °C. Then, the BrdU assay was performed following the manufacturer’s instructions. Quantitation was performed spectrophotometrically at 450 nm using a microplate spectrophotometer (Ledetect 96, Labexim Products, Lengau, Austria).

### 4.5. Cytotoxicity Assessment—LDH Assay

Optimized amounts of malignant melanoma cells (as above), normal human keratinocytes HaCaT (1 × 104/mL), and normal human melanocytes HEMa-LP (5 × 103/mL) cells were placed on 96-well plates (Nunc). The next day, the cells were washed in PBS and then exposed to increasing concentrations of FRX in fresh culture medium. The cytotoxicity was estimated based on the measurement of cytoplasmic lactate dehydrogenase (LDH) activity released from damaged cells after 72 h of exposure to FRX. The LDH assay was performed according to the manufacturer’s instruction (Cytotoxicity Detection KitPLUS LDH) (Roche). More information about the steps of the LDH assay has been described earlier [37,38].

### 4.6. Isobolographic Analysis

Isobolographic analysis is a statistical method allowing the determination of precise characteristics of pharmacodynamic interactions between drugs and chemical substances. This method is the gold standard among drug interaction assessment methods. Log-probit analysis according to Litchfield and Wilcoxon [53] was used to determine the percentage of inhibition of cell viability per concentration of fraxetin (FRX), cisplatin (CDDP), mitoxantrone (MTX), and docetaxel (DOCX) when administered singly in the FM55P, FM55M2, A375, and SK-MEL 28 cell lines measured in vitro by the MTT assay. Subsequently, from the log-probit concentration–response lines, median inhibitory concentrations (IC_50_ values) of FRX and CDDP, MTX, and DOCX were calculated. The test for parallelism between two concentration–response curves (FRX and CDDP, FRX and MTX, and FRX and DOCX) was performed as described in our previous studies [22,54,55,56]. From the experimentally denoted IC_50_ values for the drugs administered alone, median additive inhibitory concentrations for the mixture of FRX + CDDP, FRX + MTX, and FRX + DOCX at the fixed ratio of 1:1 (IC_50add_) were calculated, as described earlier [57,58]. The experimentally-derived IC_50 exp_ values for the mixture of FRX with CDDP, FRX with MTX, and FRX with DOCX (at the fixed ratio of 1:1) were determined based on the concentrations of the mixtures of CDDP or MTX or DOCX with FRX, inhibiting 50% of cell viability in the melanoma cell lines (FM55P, FM55M2, A375, and SK-MEL 28) measured in vitro by the MTT assay. Details concerning the isobolographic analysis have been published elsewhere [37,59,60].

### 4.7. Statistical Analysis

GraphPad Prism 8.0 Statistic Software was used for statistical analysis of data. The calculations were performed by one-way analysis of variance (ANOVA test) for multiple comparisons followed by Tukey’s significance test. Data are expressed as the mean ± standard error (SEM) (* *p* < 0.05, ** *p* < 0.01, *** *p* < 0.001, **** *p* < 0.0001). The IC_50_ values for FRX, CDDP, MTX, and DOCX, when administered alone, and the IC_50exp_ values for the combinations at the fixed ratio of 1:1 were calculated by computer-assisted log-probit analysis, as described elsewhere [22,53,55]. The experimentally-derived IC_50exp_ values for the mixture of FRX with CDDP, FRX with MTX, and FRX with DOCX were statistically compared with their respective theoretically additive IC_50add_ values by the use of an unpaired Student’s t-test, as reported earlier [57,58].

## 5. Conclusions

This study showed that scoparone in concentration of up to 200 µM did not have the antiproliferative activity against human melanoma. A promising natural compound is fraxetin, which significantly inhibits the growth of melanoma cell lines in vitro. The combination of fraxetin with cisplatin (CDDP) and mitoxantrone (MTX) shows an additive interaction, which seems to be a promising approach in melanoma therapy. Unfortunately, the combination of fraxetin with docetaxel may not be beneficial due to the antagonistic antiproliferative effect of both drugs used in the mixture. Further intensive experiments are needed to determine the possible mechanisms of action of the combinations of coumarins with cytostatics and to conduct animal experiments to confirm the in vivo interactions of the compounds.

## Figures and Tables

**Figure 1 ijms-24-00212-f001:**
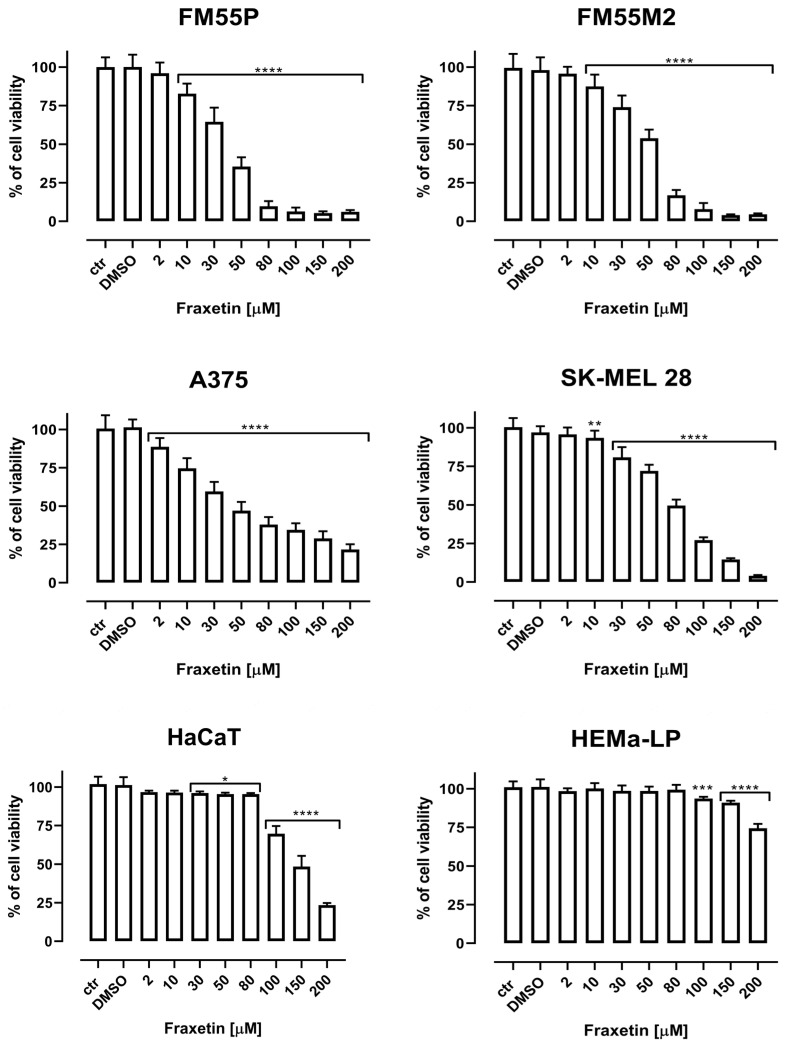
The effect of fraxetin (FRX) on the viability of malignant melanoma cancer cell lines (FM55P, FM55M2, A375, and SK-MEL 28). The normal human keratinocytes (HaCaT) and the normal human melanocytes (HEMa-LP) were measured by means of MTT assay after 72 h. Columns represent mean ± SEM (* *p* < 0.05, ** *p* < 0.01, *** *p* < 0.001, and **** *p* < 0.0001). DMSO control—0.1% DMSO in medium.

**Figure 2 ijms-24-00212-f002:**
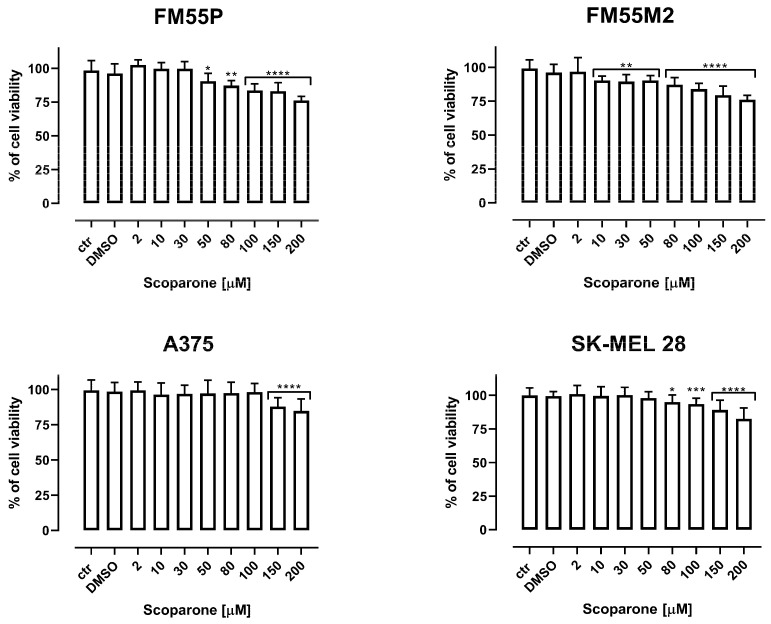
The effect of scoparone (SCP) on the viability of malignant melanoma cancer cell lines (FM55P, FM55M2, A375, and SK-MEL 28) was measured by means of MTT assay after 72 h. Columns represent mean ± SEM (* *p* < 0.05, ** *p* < 0.01, *** *p* < 0.001, and **** *p* < 0.0001). DMSO control—0.1% DMSO in medium.

**Figure 3 ijms-24-00212-f003:**
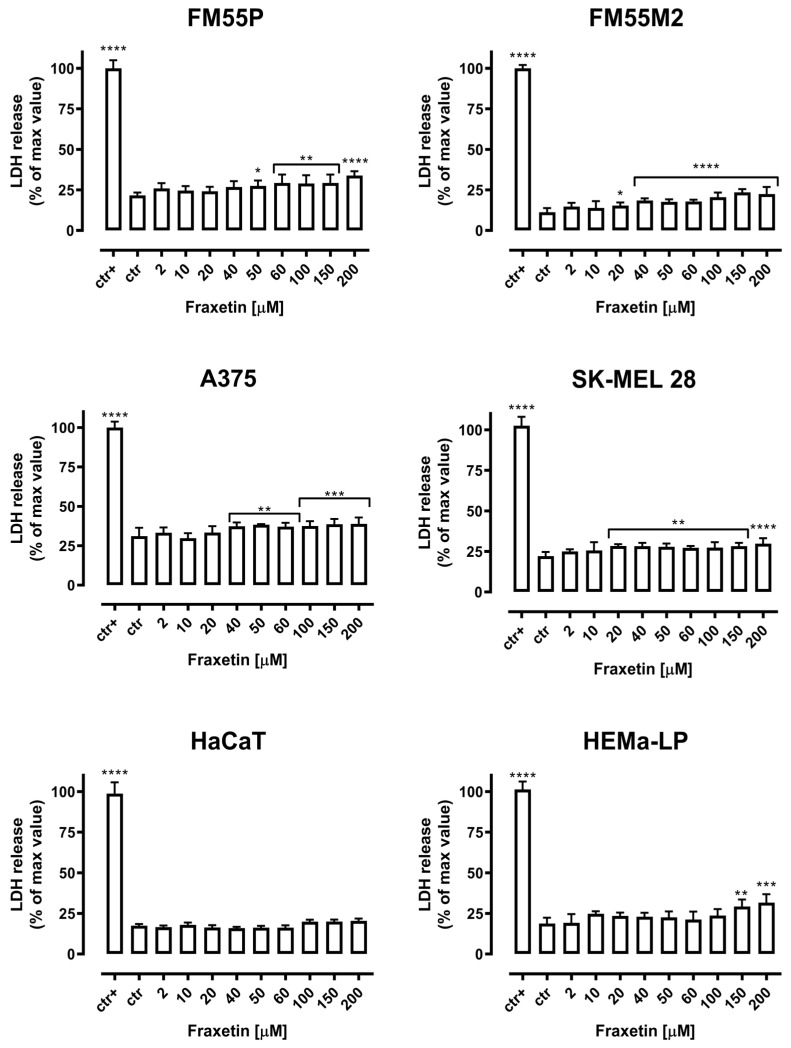
Cytotoxicity of fraxetin (FRX) to malignant melanoma cells (FM55P, FM55M2, A375, and SK-MEL 28), keratinocytes (HaCaT), and melanocytes (HEMa-LP) measured by LDH assay. Results are presented as the percentage in LDH release to the medium by treated cells versus cells grown in control medium (ctr) and cells treated with Lysis buffer (ctr+). Data are presented as mean ± SEM. * *p* < 0.05, ** *p* < 0.01, *** *p* < 0.001, and **** *p* < 0.0001.

**Figure 4 ijms-24-00212-f004:**
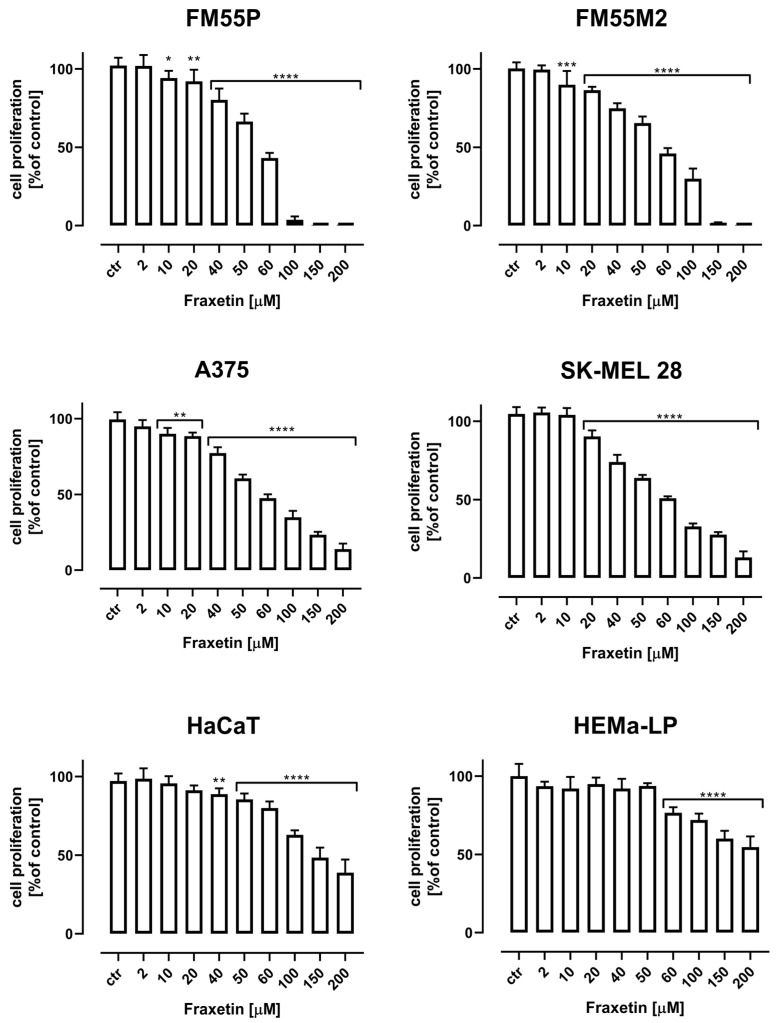
The effect of fraxetin on the proliferation of malignant melanoma cell lines (FM55P, FM55M2, A375, and SK-MEL 28), normal human keratinocytes (HaCaT), and melanocytes (HEMa-LP) was measured by BrdU assay after 72 h. The results are presented as mean ± SEM at each concentration (* *p* < 0.05, ** *p* < 0.01, *** *p* < 0.001, and **** *p* < 0.0001).

**Figure 5 ijms-24-00212-f005:**
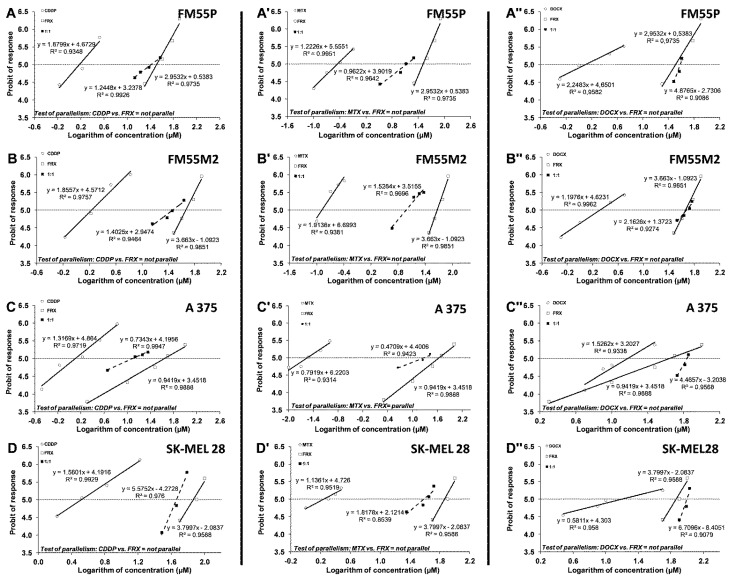
Concentration–effect lines for FRX and CDDP (**A**,**B**,**C**,**D**), FRX and MTX (**A’**,**B’**,**C’**,**D’**), and FRX and DOCX (**A’’**,**B’’**,**C’’**,**D’’**) administered alone and in combination in the fixed ratio of 1:1, illustrating the anti-proliferative effects of the drugs in the malignant melanoma cell lines: FM55P (**A**,**A’**,**A’’**), FM55M2 (**B**,**B’**,**B’’**), A375 (**C**,**C’**,**C’’**), and SK-MEL28 (**D**,**D’**,**D’’**) measured in vitro by the MTT assay.

**Figure 6 ijms-24-00212-f006:**
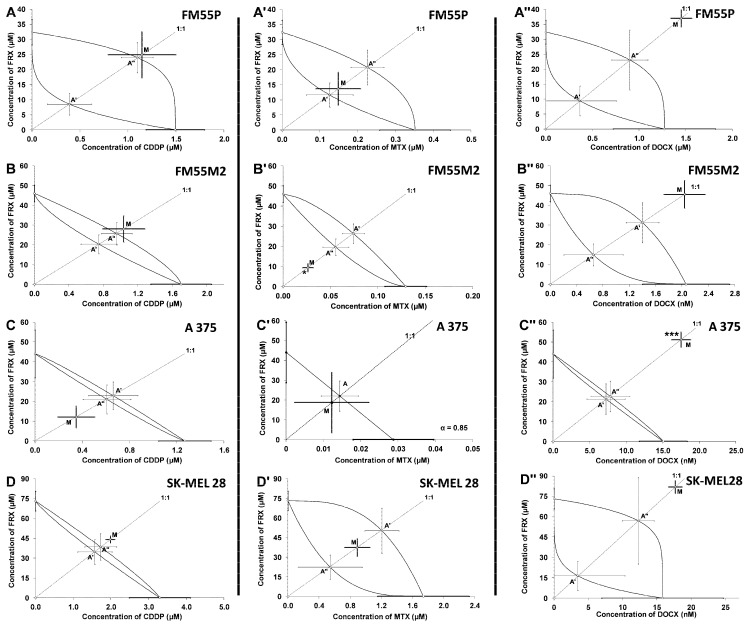
Isobolograms showing interactions between fraxetin (FRX) and cisplatin (CDDP), mitoxantrone (MTX), or docetaxel (DOCX) with respect to their anti-proliferative effects on FM55P (**A**,**A’**,**A’’**), FM55M2 (**B**,**B’**,**B’’**), A375 (**C**,**C’**,**C’’**), and SK-MEL28 (**D**,**D’**,**D’’**) malignant melanoma cell lines measured in vitro by the MTT assay. The IC_50_ ± S.E.M. for FRX and CDDP, MTX, or DOCX are plotted on the X- and Y-axes, respectively. The points A, A′, and A″ depict the theoretically calculated IC_50add_ values (±SEM). The point M on each graph represents the experimentally-derived IC_50exp_ value (±SEM) for the mixture, which produced a 50% anti-proliferative effect in the malignant melanoma cell lines. * *p* < 0.5 and *** *p* < 0.001 vs. the respective IC_50add_ value.

**Table 1 ijms-24-00212-t001:** The anti-proliferative effects of scoparone (SCP) and fraxetin (FRX) administered alone in human malignant melanoma cell lines measured in vitro by the MTT assay. Data are median inhibitory concentrations (IC_50_ values ± SEM).

Cell Line	Scoparone [µM]	Fraxetin [µM]
FM55P	>200 *	32.42 ± 4.21
FM55M2	>200 *	46.04 ± 4.17
A375	>200 *	44.03 ± 12.02
SK-MEL28	>200 *	73.16 ± 7.38

* 200 µM was the maximum possible concentration in the medium considering the 0.1% DMSO and the solubility limit of scoparone.

**Table 2 ijms-24-00212-t002:** Isobolographic analysis of interactions between FRX with CDDP, MTX, and DOCX (at the fixed ratio of 1:1) for nonparallel concentration–response effects in various malignant melanoma cell lines.

Drug Combination	Cell Line	IC_50mix_ (n_mix_) [µM]	Lower IC_50add_ (n_add_) [µM]	Upper IC_50add_ (n_add_) [µM]	Interaction
FRX + CDDP	FM55P	26.04 ± 8.02 (96)	8.51 ± 3.56 (140)	23.97 ± 4.98 (140)	Additive
	FM55M2	29.07 ± 6.88 (96)	20.34 ± 4.79 (164)	25.67 ± 5.65 (164)	Additive
	A375	12.46 ± 5.63 (96)	21.02 ± 7.26 (356)	22.91 ± 7.01 (356)	Additive
	SK-MEL28	46.05 ± 2.74 (96)	34.62 ± 9.41 (140)	38.47 ± 10.04 (140)	Additive
FRX + MTX	FM55P	13.84 ± 5.51 (72)	11.63 ± 4.01 (176)	20.78 ± 5.72 (176)	Additive
	FM55M2	9.36 ± 2.03 (96) *	19.58 ± 4.12 (192)	26.32 ± 4.83 (192)	Synergistic
	SK-MEL28	38.33 ± 7.00 (96)	22.38 ± 9.22 (140)	50.22 ± 17.27 (140)	Additive
FRX + DOCX	FM55P	38.48 ± 3.03 (96)	3.39 ± 4.93 (148)	23.09 ± 9.97 (148)	Additive
	FM55M2	47.58 ± 7.30 (96)	14.96 ± 5.49 (164)	31.26 ± 10.09 (164)	Additive
	A375	68.72 ± 5.11 (96) ***	21.35 ± 7.76 (252)	22.70 ± 7.63 (252)	Antagonistic
	SK-MEL28	99.52 ± 5.69 (72)	16.42 ± 10.66 (168)	56.95 ± 31.83 (168)	Additive

The IC_50_ values (in µM ± SEM) for the mixture of FRX with CDDP, MTX, and DOCX were determined experimentally (IC_50mix_) in four melanoma malignant cell lines in the in vitro MTT assay. The IC_50add_ values were calculated from the lower and upper isoboles of additivity. n_mix_—total number of items experimentally determined; n_add_—total number of items calculated for the additive two-drug mixture. * *p* < 0.05, *** *p* < 0.001 vs. the respective IC_50add_ value.

**Table 3 ijms-24-00212-t003:** Isobolographic analysis of interactions for parallel concentration–response effects in melanoma cell line.

Drug Combination	Cell Line	IC_50mix_ (n_mix_) [µM]	IC_50add_ (n_add_) [µM]	Interaction
FRX + MTX	A375	18.74 ± 13.21 (96)	22.03 ± 7.61 (216)	Additive

IC_50mix_—experimentally-derived IC_50_; n_mix_—number of items for experimental mixture; IC_50add_—theoretically additive IC_50_; n_add_—number of items calculated for the additive mixture.

## Data Availability

Data are contained within the article.

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
