# Peer review of "Fraxetin Interacts Additively with Cisplatin and Mitoxantrone, Antagonistically with Docetaxel in Various Human Melanoma Cell Lines—An Isobolographic Analysis"

_ijms, 2022, doi:10.3390/ijms24010212_

Round 1

Reviewer 1 Report

Paula Wró blewska-Łuczka et al. reported the inhibitory effect of coumarin compound SCP and FRX on melanoma cells when used alone, and that of FRX combined with three cytostatics (CDDP, MTX, DOCX). Further, the additive or antagonistic effects of FRX on these three cytostatics were confirmed by Isobolographic Analysis of Interactions. The manuscript was well written. However, there were some issues pending address, and a Major Revision was suggested.

Q1: In the abstract, it is mentioned that the purpose of this study was to evaluate the cytotoxic and antiproliferative effects of SCP and FRX alone and in combination with cytostatic agents (CDDP, MTX, DOCX) in vitro. But in the many texts, seemingly the part where SCP is used in combination with those agents were not mentioned. Please make proper revision about this point.

Q2: In page 3, line 106, “juman” should be “human”. There are multiple similar errors to be corrected in the manuscript.

Q3: In page 3, the part of cell viability test, the data about PBS and DMSO have no effect on cell viability should be added.

Q4: In page 5, Figure 4, the data about the effect of FRX on cell proliferation of normal melanocyte (HEMa-LP) are missing and need to be supplemented.

Q5: The resolution of Figure 6 is a bit low, and it is recommended to replace it with a clearer one.

Q6: The concentration-effect curves of the parallelism test for the combination of FRX and MTX in the A375 cell line are parallel to each other (Figure 5. C’). It is recommended to discuss the reasons in detail.

Author Response

Paula Wróblewska-Łuczka et al. reported the inhibitory effect of coumarin compound SCP and FRX on melanoma cells when used alone, and that of FRX combined with three cytostatics (CDDP, MTX, DOCX). Further, the additive or antagonistic effects of FRX on these three cytostatics were confirmed by Isobolographic Analysis of Interactions. The manuscript was well written. However, there were some issues pending address, and a Major Revision was suggested.

Q1: In the abstract, it is mentioned that the purpose of this study was to evaluate the cytotoxic and antiproliferative effects of SCP and FRX alone and in combination with cytostatic agents (CDDP, MTX, DOCX) in vitro. But in the many texts, seemingly the part where SCP is used in combination with those agents were not mentioned. Please make proper revision about this point.

A1: Due to the fact that scoparone in the range of tested concentrations (obtainable taking into account the solubility of the compound) did not show a significant inhibition of the growth of the tested melanoma cell lines (the IC50 could not be obtained), it was not possible to combine it with CDDP, MTX, DOCX. This is explained in the text in the "cell viability test" description. Additionally, such information in the Abstract has been added, as suggested.

Q2: In page 3, line 106, “juman” should be “human”. There are multiple similar errors to be corrected in the manuscript.

A2: The error has been corrected as suggested. The text has been fully proofread.

Q3: In page 3, the part of cell viability test, the data about PBS and DMSO have no effect on cell viability should be added.

A3: Results confirming no effect of 0.1% DMSO on cell proliferation have been added to the viability test graphs. PBS served as the solvent for CDDP only. The lack of its harmfulness in the tested concentrations has been presented in earlier publications, which is included in the "Material and methods".

Q4: In page 5, Figure 4, the data about the effect of FRX on cell proliferation of normal melanocyte (HEMa-LP) are missing and need to be supplemented.

A4: A graph of the effect of FRX on normal melanocyte (HEMa-LP) cell proliferation has been added to Figure 4, as suggested.

Q5: The resolution of Figure 6 is a bit low, and it is recommended to replace it with a clearer one.

A5: Both figures 5 and 6 on the printout may have poor quality, but their size is 1200 dpi, which in the online version will provide a very good quality.

Q6: The concentration-effect curves of the parallelism test for the combination of FRX and MTX in the A375 cell line are parallel to each other (Figure 5. C’). It is recommended to discuss the reasons in detail.

A6: Evaluation of parallelism of curves is an intermediate stage of isobolographic analysis. The result of evaluating the parallelism of the curves affects the appearance of the isobologram. This information has been discussed in our earlier studies.

Reviewer 2 Report

The authors have done good work by doing the isobolographic analysis in melanoma and combination of fraxetin with cisplatin and mitoxantrone may have a promising direction in melanoma therapy.

However the manuscript can be better and comprehensive by:

1. Improving the over scientific and English language.

2. Presenting and summarizing the data of anti-proliferation, IC50,  isobolographic analysis in tabular form.

Author Response

The authors have done good work by doing the isobolographic analysis in melanoma and combination of fraxetin with cisplatin and mitoxantrone may have a promising direction in melanoma therapy.

However the manuscript can be better and comprehensive by:

  1. Improving the over scientific and English language.

A1: The entire text has undergone extensive English corrections.

  1. Presenting and summarizing the data of anti-proliferation, IC50,  isobolographic analysis in tabular form.

A2: Three tables have been added: one with IC50 values for scoparone and fraxetin, and two summarizing the results of the isobolographic analysis, as suggested.

Round 2

Reviewer 1 Report

The manuscript can be accepted in the current form.